# Theranostics of Primary Prostate Cancer: Beyond PSMA and GRP-R

**DOI:** 10.3390/cancers15082345

**Published:** 2023-04-18

**Authors:** Romain Schollhammer, Marie-Laure Quintyn Ranty, Henri de Clermont Gallerande, Florine Cavelier, Ibai E. Valverde, Delphine Vimont, Elif Hindié, Clément Morgat

**Affiliations:** 1Nuclear Medicine Department, Bordeaux University Hospital, 33000 Bordeaux, France; 2INCIA, University of Bordeaux, CNRS, EPHE, UMR 5287, 33000 Bordeaux, France; 3Department of Pathology, University Hospital of Toulouse, 31000 Toulouse, France; 4Institut des Biomolécules Max Mousseron IBMM, UMR 5247, CNRS, Université Montpellier, ENSCM, Pôle Chimie Balard, 1919 Route de Mende, Cedex 5, 34293 Montpellier, France; 5Institut de Chimie Moléculaire de l’Université de Bourgogne, UMR 6302, CNRS, Université Bourgogne Franche-Comté, 9 Avenue Alain Savary, 21000 Dijon, France; 6Institut Universitaire de France (IUF), 75231 Paris, France

**Keywords:** prostate cancer, neuropeptide, PSMA, GRP-R, NTS_1_, NTS_2_, neurotensin

## Abstract

**Simple Summary:**

The accurate assessment of the aggressiveness and localization of primary prostate cancer lesions are essential for treatment decision making. Around 15% of lesions are missed by PSMA Positron-Emission tomography/computed Tomography (PET/CT). The aim of our study was to investigate the potential of novel surface markers to detect PSMA-negative lesions using immunohistochemistry and autoradiography techniques. Our work demonstrates that targeting both PSMA and neurotensin receptors might detect all intra-prostatic lesions. This new finding has implications for the future theranostics of primary prostate cancer.

**Abstract:**

The imaging of Prostate-Specific Membrane Antigen (PSMA) is now widely used at the initial staging of prostate cancers in patients with a high metastatic risk. However, its ability to detect low-grade tumor lesions is not optimal. Methods: First, we prospectively performed neurotensin receptor-1 (NTS_1_) IHC in a series of patients receiving both [^68^Ga]Ga-PSMA-617 and [^68^Ga]Ga-RM2 before prostatectomy. In this series, PSMA and GRP-R IHC were also available (n = 16). Next, we aimed at confirming the PSMA/GRP-R/NTS_1_ expression profile by retrospective autoradiography (n = 46) using a specific radiopharmaceuticals study and also aimed to decipher the expression of less-investigated targets such as NTS_2_, SST_2_ and CXCR4. Results: In the IHC study, all samples with negative PSMA staining (two patients with ISUP 2 and one with ISUP 3) were strongly positive for NTS_1_ staining. No samples were negative for all three stains—for PSMA, GRP-R or NTS_1_. In the autoradiography study, binding of [^111^In]In-PSMA-617 was high in all ISUP groups. However, some samples did not bind or bound weakly to [^111^In]In-PSMA-617 (9%). In these cases, binding of [^111^n]In-JMV 6659 and [^111^In]In-JMV 7488 towards NTS_1_ and NTS_2_ was high. Conclusions: Targeting PSMA and NTS_1_/NTS_2_ could allow for the detection of all intraprostatic lesions.

## 1. Introduction

Prostate cancer is the most common cancer in men and the third leading cause of cancer death [1]. Prostate tumors are typically multifocal, composed of a combination of cells at different stages of differentiation; the histo-prognostic grade (ISUP score) obtained from biopsy samples then guides the management. However, prostate biopsies only provide a limited representation of the intraprostatic tumoral process. Indeed, the ISUP score is frequently modified after analysis of prostatectomy specimens. In addition, it is not uncommon for biopsies to be negative, despite a strong suspicion of prostate cancer. Several studies have shown that performing multiparametric magnetic resonance imaging (mpMRI) before a series of biopsies increases the detection of lesions [2,3,4], but no imaging method is currently able to accurately estimate the histo-prognostic grade and their sensitivity is not optimal.

Focal therapies using focused ultrasound (HIFU—High Intensity Focused Ultrasound) or stereotactic radiotherapy are becoming increasingly important in the management of low-grade localized prostate cancers, mainly because of their low rates of complications. The accurate localization and characterization of the tumor lesion is therefore essential. Indeed, in many cases, no target is identified on mpMRI—thus preventing the use of these treatments. High-performance molecular imaging would guide these focal therapies.

The development of novel radiopharmaceuticals supports innovations in molecular imaging by improving sensitivity and specificity in the diagnosis and characterization of primary prostate tumors. For example, [^68^Ga]Ga-PSMA (Prostate Specific Membrane Antigen) PET/CT (Positron Emission Tomography/Computed Tomography) is now widely used at the initial staging of prostate cancers in patients with high metastatic risk and in the context of biochemical recurrence [5,6]. However, its ability to detect low-grade tumor lesions is not optimal. Novel radiopharmaceuticals with a role in this setting would be helpful.

Tissue micro-imaging is a technique that allows for the pre-clinical evaluation of radiopharmaceuticals [7,8]. We recently compared the targeting of PSMA and GRP-R (Gastrin Releasing Peptide Receptor), by means of [^111^In]In-PSMA-617 and [^111^In]In-RM2, respectively. We showed good detection of low-grade tumor lesions by [^111^In]In-RM2, superior to that of [^111^In]In-PSMA [8]. Next, we translated these results into a Phase II study using [^68^Ga]Ga-PSMA-617 PET/CT and [^68^Ga]Ga-RM2 PET/CT. Again, we demonstrated a better detection of low-grade lesions by targeting GRP-R using [^68^Ga]Ga-RM2 [9]. However, 15.6% of the lesions remained undetectable by both modalities.

New targets for prostate cancer are currently being studied, such as neurotensin receptor-1 (NTS_1_) [10], somatostatin receptor-2 (SST_2_) [11] or chemokine receptor-4 (CXCR4) [12]—suggested to be expressed in prostate cancer in a few small pilot studies. However, comparisons are needed.

Thus, the main objective of this study was to evaluate alternative targets for the better identification of intraprostatic lesions. Our strategy was based on a sequential approach: First, we prospectively performed NTS_1_ IHC in a series of patients receiving both [^68^Ga]Ga-PSMA-617 and [^68^Ga]Ga-RM2 before prostatectomy. In this series, PSMA and GRP-R IHC were also available [9]. Next, we aimed at confirming the PSMA/GRP-R/NTS_1_ expression profile by a retrospective autoradiography study and also aimed to decipher the expression of less-investigated targets such as NTS_2_, SST_2_ and CXCR4.

## 2. Materials and Methods

### 2.1. Patient Characteristics

Study 1: Formalin-fixed paraffin-embedded samples were prospectively available from patients enrolled in the NCT03604757 study, comparing [^68^Ga]Ga-PSMA-617 PET/CT to [^68^Ga]Ga-RM2 PET/CT in patients with localized prostate cancer that were candidates for radical prostatectomy. PSMA and GRP-R staining were performed during this study [9]. For the current study, 16 samples were available for additional NTS_1_ staining and comparison with GRP-R and PSMA staining (six samples were considered as non-contributors).

Study 2: Forty-six frozen samples of prostate cancer were available from the Department of Pathology of the University Hospital of Toulouse, France. Patient samples were obtained after informed consent in accordance with the Declaration of Helsinki and stored at the “CRB Cancer des Hôpitaux de Toulouse (BB-0033-00014)” collection. According to French law, the CRB Cancer collection was declared to the Ministry of Higher Education and Research (DC- 2008-463) and a transfer agreement was obtained (AC-2013-1955) after approval by ethical committees (Conseil Scientifique du Centre de Ressources Biologiques). Clinical and biological annotations of the samples were declared to the CNIL (Comité National Informatique et Libertés). None of these patients received hormone therapy or other systemic therapy prior to surgery. For each case, thirteen adjacent sections were used: one for hematoxylin-eosin-saffron (HES) staining and twelve for high-resolution microimaging (one section per radiopharmaceutical for total binding and another adjacent section for non-specific binding). An experienced pathologist manually drew tumoral areas on the HES-stained section. All patients were characterized according to their clinical and biochemical criteria including age, tumoral size (clinical and pathological sizes), PSA value and ISUP score.

### 2.2. NTS_1_—Immunohistochemistry

Immunohistochemical assessments were performed as previously described [10]. Immunohistochemistry results were expressed as an immunoreactive score (IRS) that considered staining intensity and the percentage of stained tumor cells [13]. The final IRS score (the product of the staining intensity score and the percentage of positive cells score) thus ranged from 0 to 12: IRS 0–1 means no clear expression; IRS 2–3 indicates weak expression; IRS 4–8 indicates moderate expression; IRS 9–12 indicates strong expression. In order to study associations with other parameters, IHC results were dichotomized into two groups: low expression (regrouping absent/weak expression) and high expression (regrouping moderate/strong expression).

### 2.3. Radiosynthesis and Quality Control of Radioligands

The radioligands used in this study, their respective targets and their affinities towards the target are summarized as follows: [^111^In]In-PSMA-617 targets PSMA (Ki = 2.34 ± 2.94 nM [14]), [^111^In]In-RM2 targets GRP-R (Kd = 2.9 ± 0.4 nM [15], [^111^In]In-JMV 6659 is a radioligand of NTS_1_ (Kd = 6.29 ± 1.37 nM [16]), [^111^In]In-JMV 7488 is a radioligand of NTS_2_ (Kd = 36.39 ± 4.02 nM) [17], [^177^Lu]Lu-DOTATATE targets SST_2_ (Kd = 2.0 ± 0.8 nM [18]) and [^67^Ga]Ga-pentixafor is a radioligand of CXCR4 (Kd = 24.6 ± 2.5 nM [19]). The production and control of the radiopharmaceuticals used are described in the Appendix A.

### 2.4. High-Resolution Microimaging

#### 2.4.1. Binding Assay

The protocol and recommendations edited by Reubi and co-workers for binding assays were strictly adhered to [20]. Frozen samples were kept at −80 °C. Three days before handling, samples were placed at −20 °C. The day of the experiment, samples were pre-incubated for 10 min at 37 °C in Tris-HCl buffer at pH 7.4. Then, a binding solution containing 10 nM of the radiopharmaceuticals (except [^111^In]In-JMV 7488 and [^67^Ga]Ga-pentixafor, which were used at 75 nM and 50 nM, respectively) in Tris-HCl buffer at pH 8.2, 1% of BSA (Sigma A2153), 40 μg/mL of bacitracin (Sigma^®^11,702), and 10 nM of MgCl_2_ (Sigma M8266) was applied. In order to determine the amount of non-specific binding, a large excess of cold ligand was added—more precisely, 1μM of [^nat^Ga]Ga-RM2 (Life Molecular Imaging), [^nat^Ga]Ga-PSMA-617 (ABX), neurotensin (Bachem), or [^nat^Lu]Lu-DOTATATE (ABX), 7.5 µM of levocabastine or 10 µM pentixafor were used. Samples were incubated at 37 °C for 2 h. Afterward, samples were rinsed five times for 8 min in cold Tris-HCl buffer at pH 8.2 with 0.25% of BSA, two times for 8 min in cold Tris-HCl buffer at pH 8.2 without BSA and finally, two times for 5 min in distilled water.

#### 2.4.2. Tissue Microimaging

A Beta Imager-2000 (Biospace Lab) device was used to image and quantify radioactivity in the samples. Acquisition duration was about 10 h (4 × 10^6^ counts).

### 2.5. Data Analysis

Imaging analysis was performed as previously described [7]. Briefly, HES and autoradiographic slides were fused and regions of interests (ROIs) were used to calculate the amount of specific binding. A first ROI—drawn by the pathologist to delineate tumor areas—was applied to estimate total binding, and a second ROI—corresponding to background noise—was placed around the tissue. The same ROIs were then transferred to the adjacent slice to determine non-specific binding. After subtracting background noise, specific binding (total binding—non-specific binding) was expressed as a percentage of total binding as follows: $$Specific\;binding(\%)=\frac{\left(Total\;binding\;background\right)-\left(non\;specific\;binding\;background\right)}{Total\;binding\;background}\times 100$$

### 2.6. Statistical Analysis

Data, presented as the mean ± standard deviation (SD), were compared using a non-parametric ANOVA. Statistical analyses were performed using GraphPad software (v 6.01, San Diego, CA, USA). *p* values < 0.05 were considered statistically significant. 

## 3. Results

### 3.1. Study 1: Prospective NTS_1_ IHC Study

Results are summarized in Table 1.

Immunochemistry was conducted on samples from prostatectomies of patients included in our previous in vivo study [9]. Sixteen samples were available for GRP-R, PSMA and NTS_1_ staining. Staining was cytoplasmic for PSMA and GRP-R and nuclear for NTS_1_ (Figure 1). GRP-R staining was considered positive (IRS ≥ 4) in 11 (68.8%) of 16 lesions. The median GRP-R IRS score was 4 (3–6). PSMA IRS was considered positive (IRS ≥ 4) in 15 (83.3%) of 18 lesions. The median PSMA IRS score was 11 (6–12). NTS_1_ IRS was considered positive (IRS ≥ 4) in 10 (62.5%) of 16 lesions. The median NTS_1_ IRS score was 5 (1–12).

Interestingly, all samples with negative PSMA staining (two patients with ISUP 2 and one with ISUP 3) were strongly positive for NTS_1_ staining (IRS 0 versus 12; 1 versus 12; 2 versus 12). One lesion was negative for both PSMA and GRP-R staining and strongly positive for NTS_1_ staining. On the other hand, all samples with negative NTS_1_ staining (n = 6) were positive for PSMA and five of them were positive for GRP-R. Figure 1 shows an example of a prostatic ISUP-2 sample with positive staining for NTS_1_ immunochemistry but negative staining for PSMA and GRP-R. No prostatic lesion showed negativity with all three stains for PSMA, GRP-R and NTS_1_. 

Finally, when correlating the current NTS_1_ staining results with clinical PET imaging data from patients included in the trial, four lesions were positive for NTS_1_ staining with a low [^68^Ga]Ga-PSMA-617 uptake (SUVmax < 4). One lesion was positive for NTS_1_ staining with a low [^68^Ga]Ga-RM2 uptake (Table 1).

### 3.2. Study 2: Retrospective Study of the Expression of PSMA, GRP-R, NTS_1_, NTS_2_, SST_2_ and CXCR4 on Samples of Primary Prostate Cancer

Patient characteristics were summarized in Table 2.

### 3.3. Radiopharmaceuticals

[^111^In]In-RM2 was used at 3.9 GBq/µmol, [^111^In]In-PSMA-617 was used at 10.0 GBq/µmol, [^111^In]In-JMV 6659 was used at 2.2 GBq/µmol, [^111^In]In-JMV 7488 was used at 3.4 GBq/µmol, [^67^Ga]Ga-pentixafor was used at 0.3 GBq/µmol and [^177^Lu]Lu-DOTATATE was used at 14.9 GBq/µmol. All radiopharmaceuticals were produced at radiochemical purity > 95%.

### 3.4. Quantitative Analysis

The specific binding (expressed as percentage over total binding) of each radiopharmaceutical according to its ISUP score is shown in Table 3. 

[^111^In]In-PSMA-617 binding was significantly higher than [^111^In]In-RM2, [^111^In]In-JMV 6659, [^111^In]In-JMV 7488 and [^177^Lu]Lu-DOTATATE for all ISUP scores (*p* < 0.0001). Interestingly, no significant difference was found between [^111^In]In-PSMA-617 and [^67^Ga]Ga-pentixafor, but the numbers of samples that could be assessed for CXCR4 was more limited.

For each radiopharmaceutical, there was no significant difference in binding intensity between various ISUP scores.

Overall, binding of [^111^In]In-PSMA-617 was high in all ISUP groups. However, it was interesting to see that some samples did not bind or bound weakly [^111^In]In-PSMA-617 (9%). Therefore, a search for novel targets is needed. Below, we report the number of samples for which the binding intensity of the radiopharmaceutical was at least equal to that of [^111^In]In-PSMA-617 (Table 4), six for [^111^In]In-JMV 6659 (1 ISUP-1, 2 ISUP-3 and 3 ISUP-4), four for [^111^In]In-JMV 7489 (1 ISUP-1, 1 ISUP-2, 1 ISUP-3 and 1 ISUP-5), three for [^177^Lu]Lu-DOTATATE (2 ISUP-1 and 1 ISUP-3), three for [^67^Ga]Ga-pentixafor (2 ISUP-1 and 1 ISUP-5) and two for [^111^In]In-RM2 (1 ISUP-1 and 1 ISUP-5).

The number of samples for which the specific binding of a radiopharmaceutical was equal or higher than [^111^In]In-RM2 is reported in Table 5: forty-three for PSMA (7 ISUP-1, 11 ISUP-2, 8 ISUP-3, 8 ISUP-4 and 9 ISUP-5), twenty for NTS_2_ (2 ISUP-1, 5 ISUP-2, 5 ISUP-3, 5 ISUP-4 and 3 ISUP-5), nineteen for SST_2_ (4 ISUP-1, 6 ISUP-2, 3 ISUP-3, 2 ISUP-4 and 4 ISUP-5) eight for NTS_1_ (2 ISUP-1, 1 ISUP-2, 3 ISUP-3 and 2 ISUP-4) and eight for CXCR4 (4 ISUP-1, 2 ISUP-2 and 2 ISUP-5).

One ISUP-2 sample with low binding of [^111^In]In-PSMA-617 and negative binding for [^111^In]In-RM2 was positive only for NTS_2_.

One ISUP-5 sample with negative binding of [^111^In]In-PSMA-617 and [^111^In]In-RM2 was positive for SST_2_, NTS_2_ and CXCR4.

To illustrate these results, three different cases are presented in Figure 2, Figure 3 and Figure 4. 

## 4. Discussion

Several radiopharmaceuticals have been developed to help in the staging of prostate cancer. The radiolabeled analog of the essential amino acid leucine ^18^F-FACBC (^18^F-Flucicovine) does not demonstrate high specificity for imaging in primary prostate cancer [21]. Furthermore, ^11^C-Acetate—marking lipid metabolism—cannot reliably distinguish benign prostatic hyperplasia from prostate tumors. Finally, ^11^C/^18^F-Choline—another marker of lipid metabolism—shows lower sensitivity than mpMRI for the detection of primary prostate cancer [22]. Thus, the search for novel targets appears necessary for the initial assessment of the aggressiveness of primary prostate tumors.

PSMA and GRP-R have been investigated for the initial staging of prostate cancer. In a prospective study enrolling 56 intermediate grade prostate cancer patients before prostatectomy, PSMA PET was found to be accurate in detecting intraprostatic lesions of ISUP ≥ 2. Contrarily, the detection rate of PSMA PET was low for ISUP-1 lesions. Touijer et al. prospectively investigated [^68^Ga]Ga-RM2 PET/CT in 16 patients before radical prostatectomy; the performance of [^68^Ga]Ga-RM2 PET/CT imaging did not significantly differ compared to mpMRI in terms of sensitivity, specificity or accuracy [23].

Our previous study showed similar findings, as [^68^Ga]Ga-PSMA-617 PET/CT was useful for depicting higher ISUP score lesions and [^68^Ga]Ga-RM2 PET/CT had a higher detection rate for low-ISUP tumors [9]. In the lesion-based analysis (including lesions < 0.1 cc), [^68^Ga]Ga-PSMA-617 PET/CT detected 74.3% of all tumor lesions and [^68^Ga]Ga-RM2 PET/CT detected 78.1%. However, paired examinations showed negative uptake in 15.6% of lesions by both modalities. Therefore, the objective of this work was to explore new targets to detect these unseen lesions.

The prospective immunochemistry study performed in this work confirms the interest in NTS_1_, as all PSMA negative lesions were strongly positive for NTS_1_. Moreover, all negative NTS_1_ staining lesions (37.5%) were positive for PSMA staining and positive for GRP-R staining in five patients (31%). Our results consolidate a previous study demonstrating that PSMA-negative samples from Gleason scores of 5, 6 or 7 were all NTS_1_-positive [24]. Thus, the interest in NTS_1_ might be greater than for GRP-R in low histological grade tumors, but comparison with GRP-R is obviously needed. Unfortunately, no NTS_1_ imaging radiopharmaceutical has yet shown interesting results when applied to humans [25]. Work is ongoing to find stabilized NTS_1_ analogues suitable for imaging [16,26]. These new data should also be considered with caution as IHC results do not necessarily translate into similar findings in vivo.

With this in mind, we performed a retrospective micro-imaging study comparing PSMA, GRP-R, NTS_1_ as well as NTS_2_, SST_2_ and CXCR4 expression using specific radiopharmaceuticals that would be more representative of in vivo behavior. Overexpression of the NTS_2_ receptor in prostate cancer has been reported; an in vitro study has assessed the potential use of the NTS_2_ receptor as a target by analyzing its expression patterns in human prostate cell lines and primary prostate cell cultures—NTS_2_ was found in cells with luminal phenotype [27]. Other studies are needed to confirm these results. SST_2_ is also overexpressed in prostate cancer—especially in cases of neuroendocrine differentiation [11,28]. CXCR4 overexpression has also been reported in prostate cancer; studies have shown that CXCR4 is a key regulator of tumor dissemination [12]. An in vitro study comparing adjacent normal endothelial cells to prostate tumor vasculature highlighted CXCR4 as a potential novel target to interfere with prostate cancer angiogenesis [29].

While our work shows the superiority of PSMA for the detection of intraprostatic lesions, with a significant higher binding of [^111^In]In-PSMA-617 than [^111^In]In-RM2, [^111^In]In-JMV 6659, [^177^Lu]Lu-DOTATATE or [^111^In]In-JMV 7488 for all ISUP-score groups (no significant difference was found for CXCR4—mostly due to a lack of power), and PSMA PET has now entered into guidelines [30], alternative targets are necessary in the event of PSMA negativity. In a previous study enrolling fifty newly diagnosed patient with prostate cancer, the [^68^Ga]Ga-PSMA-617 PET/CT was negative in 12.5% [31]. Targeting the GRP-R is expected to cover the limitations of PSMA [9]. In our work, in ISUP scores 1, binding of [^111^In]In-RM2, [^111^In]In-JMV 6659 and [^111^In]In-JMV 7488 were higher than that of [^111^In]In-PSMA-617 in one case (the same case for [^111^In]In-RM2 and [^111^In]In-JMV 7488, a different one for [^111^In]In-JMV 6659). In the ISUP-2 group, only [^111^In]In-JMV 7488 showed a higher signal than [^111^In]In-PSMA-617. In the ISUP 3 group, two samples showed higher [^111^In]In-JMV 6659 binding than [^111^In]In-PSMA-617—one sample showed higher binding of [^177^Lu]Lu-DOTATATE than [^111^In]In-PSMA-617 and another sample also showed higher binding of [^111^In]In-JMV 7488 than [^111^In]In-PSMA-617. Indeed, no statistical analyses were performed due to the low number of samples. 

Overall, the most interesting targets to cover PSMA-negative lesions appear to be NTS_1_ and NTS_2_—with, respectively, four and six cases with superior or equivalent detection than PSMA, covering all ISUP scores. It is interesting to note that combining PSMA and NTS_1_/NTS_2_ could allow for the detection of all intraprostatic lesions. The new findings in this work also highlight the potential of multireceptor-targeting radioprobes that can still bind one target (NTS_1_ or NTS_2_ or GRP-R) when the other is lost (PSMA). Works are ongoing to optimize radiolabeled PSMA/GRP-R heterobivalent probes [32], while the development of PSMA/NTS_1_ heterodimers has only been described once [33]. Overall, this work sheds light on the abundance of different neuropeptide receptors (mainly neurotensin receptors) in different physiopathological states of prostate cancer. 

The improved detection of lesions allows for better mapping of prostate tumor pathology, which is necessary for biopsy guiding to decrease the discordance rate of staging of biopsies and final staging of prostatectomy samples. Finally, the possibility of a more precise detection and characterization of intra-prostatic lesions opens new avenues for radiotherapy planning and/or focal treatments.

The reader should be aware that it was not our aim to compare radiopharmaceuticals, but rather to use them to quantify receptor density in primary prostate cancer samples. Moreover, in this work, we were not able to provide the uptake (as a percentage of the applied dose) of each radiopharmaceutical.

## 5. Conclusions

In this work, we have compared GRP-R, PSMA, NTS_1_, NTS_2_, SST_2_ and CXCR4 expression in vitro in primary prostate cancer samples. Our results confirm that PSMA remains the best target in tumor detection at initial staging—especially for high grade lesions. Interestingly, targeting NTS_1_ and NTS_2_ allowed us to detect all PSMA-negative lesions more precisely than GRP-R in vitro. Future in vivo prospective studies must confirm these data.

## Figures and Tables

**Figure 1 cancers-15-02345-f001:**
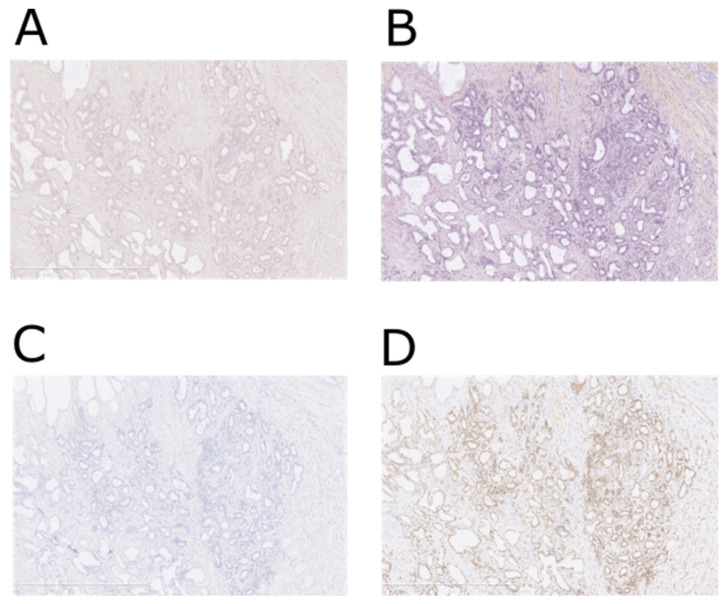
Example of a prostatic ISUP-2 sample (HES staining in (**A**)) negative for PSMA (**B**) and GRP-R (**C**) immunohistochemistry, but with positive staining for NTS_1_ immunochemistry (**D**). Images were taken at 10× magnification.

**Figure 2 cancers-15-02345-f002:**
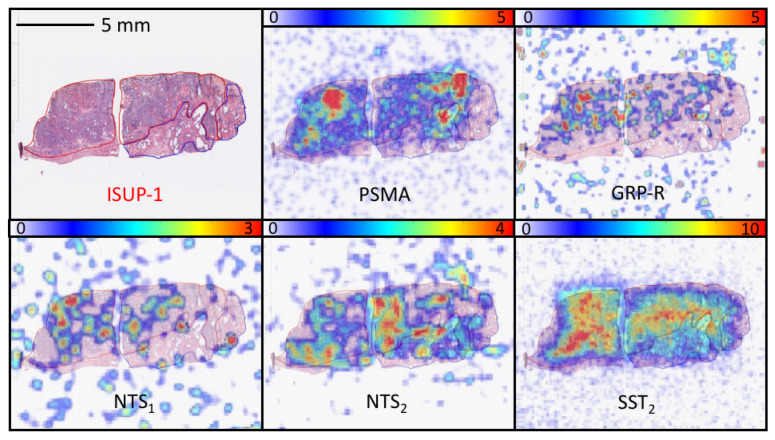
Comparison of the binding between PSMA, GRP-R, NTS_1_, NTS_2_ and SST_2_-specific radiopharmaceuticals on an ISUP-1 sample. The red line drawing on the HES slice corresponds to the tumor area. Specific binding = 92% for PSMA, 90% for GRP-R, 67% for NTS_1_, 92% for NTS_2_ and 47% for SST_2_. Color scale refers to cps/mm².

**Figure 3 cancers-15-02345-f003:**
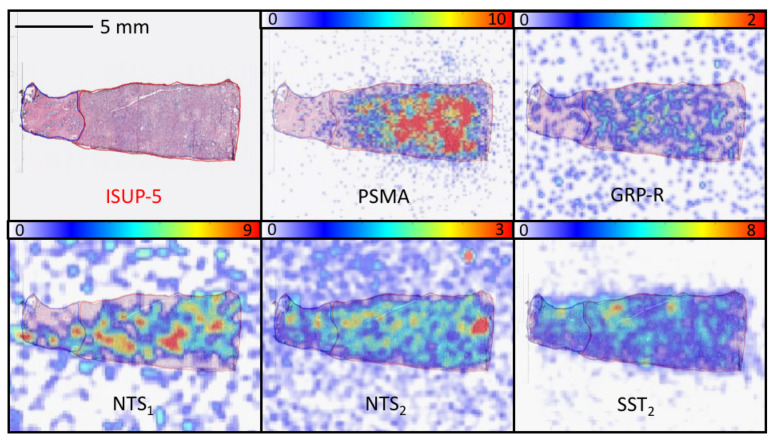
Comparison of the binding between PSMA, GRP-R, NTS_1_, NTS_2_ and SST_2_-specific radiopharmaceuticals on an ISUP-5 sample. The red line drawing on HES slice corresponds to the tumor area. Specific binding = 45% for PSMA, 40% for NTS_1_, 95% for NTS_2_ and 40% for SST_2_. Specific binding was not available for GRP-R due to technical issues. Color scale refers to cps/mm².

**Figure 4 cancers-15-02345-f004:**
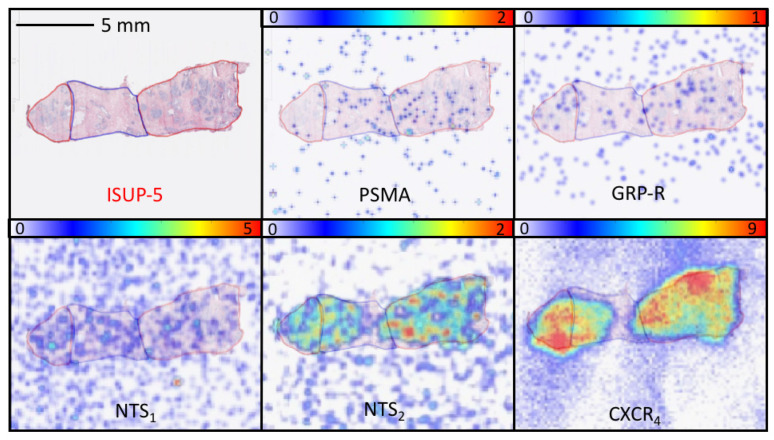
Comparison between PSMA, GRP-R, NTS_1_, NTS_2_ and CXCR4-specific radiopharmaceuticals on an ISUP-5 sample. The red line drawing on HES slice corresponds to the tumor area. PSMA, GRP-R and NTS_1_ samples showed no uptake on the tumoral area. Contrarily, NTS_2_, and CXCR4 showed a strong tumor uptake. Specific binding = 0% for PSMA, 0% for GRP-R, 0% for NTS_1_, 71% for NTS_2_ and 100% for CXCR4. Color scale refers to cps/mm².

**Table 1 cancers-15-02345-t001:** PSMA, GRP-R and NTS_1_ immunochemistry staining with IRS score according to uptake intensity (Standard Uptake Value—SUVmax) of [^68^Ga]Ga-PSMA-617 and [^68^Ga]Ga-RM2 Positron Emission Tomography (PET) imaging.

		PSMA		GRP-R		NTS_1_
Patient	ISUP Score	IRS	SUVmax	IRS	SUVmax	IRS
1	1	6	2.8	6	4.8	6
2	2	9	4.5	3	5.1	6
3	2	9	4.7	8	6.3	0
4	2	0	5	4	5.3	12
5	2	1	3.4	1	7.5	12
6	3	12	6.8	4	8.3	1
7	3	2	3.6	8	8.9	12
8	4	9	2.8	6	2.4	3
9	4	9	8.5	6	2.8	2
10	5	12	13.3	4	7.5	2
11	5	12	5.9	4	7.2	4
12	5	12	12.5	2	2.8	8
13	5	6	7.1	1	9.1	4
14	5	12	3.7	4	10.5	12
15	5	12	7.8	4	9	6
16	5	12	20.4	2	3.7	1

**Table 2 cancers-15-02345-t002:** Characteristics of the patients from which samples have used in this study. ND not determined. PSA prostate specific antigen. * All patients were NxMX or N0M0 except for patient no. 31, who was stage NxM1.

Patient	Age	ISUP	Gleason Score	PSA (ng/mL)	Clinical Tumoral Size: cT	Pathological Tumoral Size: pT	Metastatic Risk
1	67	1	6 (3 + 3)	3.7	1	2c	High
2	65	1	6 (3 + 3)	5.26	1	2c	High
3	57	1	6 (3 + 3)	4.38	1	2a	Low
4	51	1	6 (3 + 3)	3.7	2	2a	Low
5	63	1	6 (3 + 3)	10	1	2c	High
6	48	1	6 (3 + 3)	4.51	1	2c	High
7	56	1	6 (3 + 3)	4.4	2	2c	High
8	55	1	6 (3 + 3)	3.7	2	2c	High
9	70	2	7 (3 + 4)	10.5	1	3a	High
10	67	2	7 (3 + 4)	5.65	2	2c	High
11	57	2	7 (3 + 4)	6	1	3a	High
12	66	2	7 (3 + 4)	10	2	2c	High
13	59	2	7 (3 + 4)	13	2	2b	Intermediate
14	66	2	7 (3 + 4)	14	2	2c	High
15	67	2	7 (3 + 4)	14	1	3a	High
16	66	2	7 (3 + 4)	10.4	0	3a	High
17	67	2	7 (3 + 4)	12.5	1	3a	High
18	55	2	7 (3 + 4)	13	1	3a	High
19	49	2	7 (3 + 4)	14.28	2	3b	High
20	64	3	7 (4 + 3)	8	1	3a	High
21	60	3	7 (4 + 3)	5.67	1	3b	High
22	66	3	7 (4 + 3)	4.28	2	3a	High
23	58	3	7 (4 + 3)	7.6	2	3a	High
24	71	3	7 (4 + 3)	6.4	2	3a	High
25	67	3	7 (4 + 3)	7.6	2	2c	High
26	63	3	7 (4 + 3)	28	2	nd	High
27	63	3	7 (4 + 3)	25.6	3	3b	High
28	68	3	7 (4 + 3)	19	2	3a	High
29	53	3	7 (4 + 3)	20	2	3a	High
30	75	4	8 (4 + 4)	6	3	1b	High
31 *	71	4	8 (4 + 4)	285	4	nd	High
32	63	4	8 (4 + 4)	7	2	3a	High
33	70	4	8 (4 + 4)	3.9	2	3a	High
34	70	4	8 (4 + 4)	9.95	1	2c	High
35	74	4	8 (5 + 3)	nd	nd	nd	High
36	66	4	8 (4 + 4)	44	2	3a	High
37	59	4	8 (4 + 4)	14	2	4	High
38	73	5	9 (4 + 5)	10	nd	3b	High
39	72	5	9 (4 + 5)	20	3	3b	High
40	63	5	9 (4 + 5)	27	3	3b	High
41	54	5	9 (4 + 5)	30	3	3a	High
42	60	5	9 (4 + 5)	12.6	2	3a	High
43	66	5	9 (4 + 5)	4.4	2	2a	High
44	63	5	9 (5 + 4)	5	2	3a	High
45	70	5	9 (4 + 5)	24.5	2	3b	High
46	56	5	9 (4 + 5)	26	3	3a	High

**Table 3 cancers-15-02345-t003:** ISUP-based stratification and statistical analysis of samples for each target. Specific binding % ± standard deviation (number of samples). Non-parametric one-way ANOVA (Kruskal–Wallis test). *p* < 0.05 was considered significant. * stands for significant difference.

ISUP	PSMA	GRP-R	NTS_1_	SST_2_	NTS_2_	CXCR_4_
1	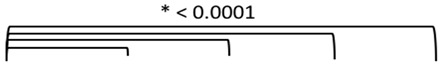
78.8%±10.0 (8)	44.7%±51.3 (8)	38.3%±33.6 (6)	43.5%±33.0 (8)	29.4%±37.5 (5)	62.4%±21.0 (4)
2	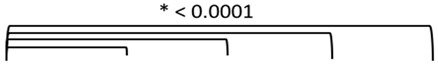
81.0%±15.4 (11)	8.2%±18.3 (11)	10.0%±20.0 (4)	34.6%±28.6 (9)	37.1%±29.6 (7)	36.4%±3.1 (2)
3	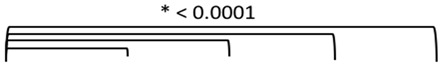
89.0%±9.8 (8)	27.5%±29.7 (9)	64.7%±68.7 (6)	77.6%±147 (9)	36.4%±38.9 (9)	0%±0 (0)
4	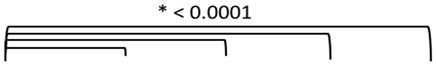
94.7%±4.6 (8)	16.2%±27.8 (8)	39.1%±47.2 (7)	7.4%±10.0 (3)	37.3%±32.6 (6)	32.0%±45.2 (2)
5	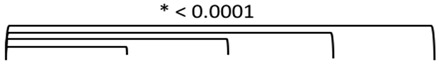
73.6%±25.7 (8)	20.8%±39.4 (8)	13.3%±23.0 (7)	32.0%±28.7 (9)	54.6%±35.6 (6)	43.9%±73.3 (4)
Total	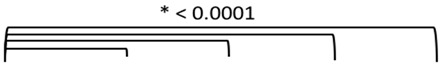
83.3%±16.0 (43)	22.5%±34.7 (44)	34.2%±44.8 (30)	43.9%±75.3 (38)	39.0%±33.8 (33)	46.8%±43.9 (12)

**Table 4 cancers-15-02345-t004:** Number of Delta ≥ PSMA for GRP-R, NTS_1_, SST_2_, NTS_2_ and CXCR_4_. Note: ISUP5 NTS_2_ >> PSMA (95% vs. 44%).

ISUP	GRP-R	NTS_1_	SST_2_	NTS_2_	CXCR4
1	1	1	2	1	2
2	0	0	0	1	0
3	0	2	1	1	0
4	0	3	0	0	0
5	1	0	0	1	1
Total	2	6	3	4	3

**Table 5 cancers-15-02345-t005:** Number of specific binding ≥ GRPR for PSMA, NTS_1_, SST_2_, NTS_2_ and CXCR_4_.

ISUP	PSMA	NTS_1_	SST_2_	NTS_2_	CXCR4
1	7	2	4	2	4
2	11	1	6	5	2
3	8	3	3	5	NA
4	8	2	2	5	NA
5	9	0	4	3	2
Total	43	8	19	20	8

## Data Availability

The data presented in this study are available on request from the corresponding author. The data are not publicly available due to privacy.

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
