# Peer review of "Theranostics of Primary Prostate Cancer: Beyond PSMA and GRP-R"

_cancers, 2023, doi:10.3390/cancers15082345_

Round 1
Reviewer 1 Report
This study evaluated the role of theranostics in prostate cancer beyond the PSMA and GRP-R. Summarized that targeting the PSMA and NTS1/NTS2 it could be possible detect all intraprostatic lesions. Length and readability are good. The choice of the topic is intresting because the article investigates novel surface markers to detect PSMA negative lesions in primary diagnostic setting. The study conducted well and the appropriate statistic tests were performed. Te results are presented in a logical manner and the discussion flows well. The manuscript is well organized and written.
Limitation of the study is the small number of samples examined and that these findings do not necessarily trans;ate in similar results in vivo. Moreover nowadays there is alot of converasation regarding the necessity of diagnose and treat low grade prostate tumors. On the other hand I believe that the article could contribute to the necessity of reduction of the downstaging we all see after radical prostatectomy in patients with prostate cancer
In my opinion this paper shoud be accepted without any revision
Author Response
This study evaluated the role of theranostics in prostate cancer beyond the PSMA and GRP-R. Summarized that targeting the PSMA and NTS1/NTS2 it could be possible detect all intraprostatic lesions. Length and readability are good. The choice of the topic is intresting because the article investigates novel surface markers to detect PSMA negative lesions in primary diagnostic setting. The study conducted well and the appropriate statistic tests were performed. Te results are presented in a logical manner and the discussion flows well. The manuscript is well organized and written.
Limitation of the study is the small number of samples examined and that these findings do not necessarily trans;ate in similar results in vivo. Moreover nowadays there is alot of converasation regarding the necessity of diagnose and treat low grade prostate tumors. On the other hand I believe that the article could contribute to the necessity of reduction of the downstaging we all see after radical prostatectomy in patients with prostate cancer
In my opinion this paper shoud be accepted without any revision
We thank the reviewer for the positive feedback regarding our work
Reviewer 2 Report
This is a very interesting manuscript that describes the limitations of current radiopharmaceutical preparations used for diagnosis of primary and secondary prostate cancer lesions. The manuscript is well-written and quite detailed. The tables presented herein are a bit overwhelming and might be considered discouraging towards the readership of Cancers. Second to this, the authors might describe novel approaches towards usage of new PSMA/GRPR-targeting heterodimers for prostate cancer when one or the other radiopharmaceutical might not be adequeate for detection of lesions. The reviewer is unaware of new heterodimers that also include NT-targeting agents but that would also be useful to describe if those agents are described in the literature.
Author Response
We thank the reviewer for his positive opinion and supporting suggestions for improving our manuscript. We introduced a novel Table 3 that should be more readable. The reviewer’s remark concerning radiolabeled heterodimers is absolutely right. We have added few words and references regarding this concept. We have also find a reference of PSMA/NT heterodimer that fits well with the results of our work (Development of Bispecific NT-PSMA Heterodimer for Prostate Cancer Imaging: A Potential Approach to Address Tumor Heterogeneity. Ma X, Wang M, Wang H, Zhang T, Wu Z, Sutton MV, Popik VV, Jiang G, Li Z. Bioconjug Chem). We also would like to advise the reviewer that we have in hand a PSMA/NT heterodimer. This compound is in a very early state of development, but we hope we will be able to share results on this new compound soon.
Reviewer 3 Report
This is a very interesting paper with an interesting methodological choice. However as a physicist working a lot with autoradiography I feel slightly confused by how the data was actually analysed. There is a reference to a 2019 paper which shows how SNR and Delta are calculated. But in neither that paper nor the current one there seems to be any connection to how much activity the sample was incubated with? The most obvious way to calculate uptake in an autoradiography image would be to calibrate the instrument to convert counts to Bq and then express the results as percentage of the activity added during incubation, of course with corrections for the decay time. Is the “specific binding” simply the percentage of uptake that was in the tumor ROI compared to in the whole tissue section? If so that introduces a problem when you are not using the same tissue section for all measurement. Even if adjacent, the sections are not identical and the mechanical procedure of sectioning will also stretch and deform each section differently. Considering this there should at least be some discussion about why not more measurements were done on a single section using multi-isotope imaging which the Betaimager instrument should be capable of.
So overall I feel that the methodology of reaching the results from the autoradiography must be much more clearly explained for this paper to be accepted. It is fine to reference the 2019 paper but you must at least have a brief and, most importantly, clear, explanation in this paper.
There are also several mentions of the “Supplementary material” but no such material was provided for review.
What do the scale bars in Figure 2-4 indicate?
Author Response
This is a very interesting paper with an interesting methodological choice. However as a physicist working a lot with autoradiography I feel slightly confused by how the data was actually analysed. There is a reference to a 2019 paper which shows how SNR and Delta are calculated. But in neither that paper nor the current one there seems to be any connection to how much activity the sample was incubated with? The most obvious way to calculate uptake in an autoradiography image would be to calibrate the instrument to convert counts to Bq and then express the results as percentage of the activity added during incubation, of course with corrections for the decay time. Is the “specific binding” simply the percentage of uptake that was in the tumor ROI compared to in the whole tissue section? If so that introduces a problem when you are not using the same tissue section for all measurement. Even if adjacent, the sections are not identical and the mechanical procedure of sectioning will also stretch and deform each section differently. Considering this there should at least be some discussion about why not more measurements were done on a single section using multi-isotope imaging which the Betaimager instrument should be capable of.
So overall I feel that the methodology of reaching the results from the autoradiography must be much more clearly explained for this paper to be accepted. It is fine to reference the 2019 paper but you must at least have a brief and, most importantly, clear, explanation in this paper.
The points raised by the reviewer are important. Indeed, as user of autoradiography, the possibility to convert cps to activity is appealing. Therefore, we worked a lot to find a way to accurately perform a calibration curve. We tried gels, engrave slides, TLC, etc., but the results obtained were not satisfactory. The point is that the BetaImager device needs high voltage but also TEA gas. Gels and TLC are not fully compatible with high voltage due to fragility and created artifacts. The TEA gas is very sensitive to any modification of height so the use of engraved slides is challenging because the voltage will be different between the top and the bottom of the slides. Considering the specific binding, it was calculated by using information from two adjacent slices (displacement). The first slice was incubated with the radiopharmaceutical alone (ex: 111In-PSMA giving access to total binding) and the adjacent slices was co-incubated with the same amount of 111In-PSMA + a large excess of a non-radioactive PSMA inhibitor (giving access to non specific binding). Next specific binding is calculated by (total binding – background) - (non specific binding – background) / (total binding – background). By using this methodology, multi-isotope imaging is not feasible (please note that the BetaImager is capable of multi-isotope imaging) because the same radionuclide was used for total and non specific binding. We are not aware of the possibility of performing displacement study in a single slide.
Overall, we provide now more details regarding the methodology of how images were analyzed and also added a caveat sentence in the discussion regarding the absence of calibration to provide % of applied dose (please note that, as stated at the end of the manuscript, the objective of this study was to compare receptors expression not radiopharmaceuticals).
There are also several mentions of the “Supplementary material” but no such material was provided for review. We apologize for this oversight. The corresponding file has been submitted with the article during the submission process but was not shared with the reviewer. We provide again the file. Thank you
What do the scale bars in Figure 2-4 indicate? It refers to cps/mm². This information has been added to all figures’ caption. Thank you